# Decoding fertility behaviour of married women in Zambia: A multipronged analysis of bio-demographic, socio-economic and intermediate factors

**Bwalya Bupe Bwalya** [1,2*], **Clifford Odimegwu** [2]

**1** Department of Economics, School of Social Sciences, Mulungushi University, Kabwe, Zambia,
**2** Demography and Population Studies Programme, Schools of Public Health and Social Sciences, University of Witwatersrand, Johannesburg, South Africa

\* bwalya1983@gmail.com

## Abstract

Despite national efforts and a decrease in total fertility rate, Zambia's fertility remains high compared to global and regional averages. While previous research has examined the influence of bio-demographic and socio-economic factors, it has neglected the role of overlooked intermediate variables such as age at first marriage, contraception use, and abortion. This study investigated the influence of these variables, while controlling for bio-demographic and socio-economic factors, on women's fertility behaviour in Zambia. The study drew upon data from three cross-sectional Zambia Demographic Health Surveys (2007, 2013–14, 2018) to investigate fertility behaviour among 18,299 married women, measured by the number of children ever born (CEB). Descriptive and inferential statistical analyses, including Analysis of Variance and Negative Binomial regression, were conducted using Stata 14.2 to identify factors associated with women's fertility behaviour. The regression results are presented as adjusted incidence rate ratios with confidence intervals. Married Zambian women's fertility behaviours, as measured by CEB are concerning, and exhibit patterns influenced by intermediate factors like age at first marriage and abortion. Early marriage, specifically before the age of 18, is associated with higher likelihood of having more CEB than those who married at 18 years or above (AIRR = 1.10, 95% CI: 1.07–1.14 in 2007; AIRR = 1.10, 95% CI: 1.10–1.12 in 2013; and AIRR = 1.07, 95% CI: 1.05–1.10). Conversely, married women who reported having had an abortion were more likely to have fewer CEB. In 2018, women with a history of abortion had lower likelihood of higher CEB compared to those without (AIRR = 0.94, 95% CI: 0.91–0.97). Besides, demographic and socio-economic factors such as age, education level, geographic location, and decision-making autonomy were found to significantly impact women's fertility behaviour. This study shows that the two intermediate variables of age at first marriage and a history of prior abortion are more influential on women fertility behaviour than contraception among married women in Zambia. To effectively address stalled fertility and improve women's reproductive health, policies should address early marriage, enhance access to sexual reproductive healthcare, and empower women to enable them make informed reproductive decision-making.

**Data availability statement:** The data used in this study were obtained from the Demographic and Health Surveys (DHS) Program. The DHS Program provides public access to these datasets upon a simple registration process through their website: https://dhsprogram. com/data/available-datasets.cfm.

**Funding:** The authors received no specific funding for this work.

**Competing interests:** The authors have declared that no competing interests exist.

## Introduction

Globally, the population was estimated to have reached 8 billion in 2022, marking a 1 billion increase from the previous year [1]. The increase maybe partially attributed to undesirable fertility behaviours (FBs) in regions like sub-Saharan Africa (SSA), particularly Zambia [2,3]. Fertility behaviour (FB) refers to a woman's patterns of childbirth, number and timing of births as well as the establishment of a marital union and the use of contraceptives [4,5]. Though SSA has experienced a reduction in the total fertility rate (TFR) over the past 30 years, it remains twice the world average of 4.6 [6]. To achieve significant progress in addressing poor maternal and child health outcomes like anaemia, morbidity, and mortality, it is important to understand the drivers of such FB in this region [7,8].

Zambia, a nation within the SSA region, is characterized by a high fertility rate. Despite a decline in the Total Fertility Rate (TFR) from 6.5 in 1992 to 4.7 in 2018, the reduction has been slow, a rate that is greater than the estimate for the entire SSA [6]. This observed TFR can be attributed to several underlying factors and FBs like early sexual debut with a median age at marriage of 19.1 years, a significant teenage pregnancy rate of 29%, and low modern contraception use at 48% [9]. Furthermore, the population growth rate has increased from 2.8% between 2000 and 2010 to 3.4% between 2010 and 2022 [10].

Various studies conducted in Zambia have explored factors associated influencing fertility behaviours [11–13]. These investigations have collectively identified various bio-demographic, socio-economic, and cultural factors associated with fertility behaviour such as early child-bearing, desired and preferred number of children, exposure to family planning, economic empowerment and household wealth to mention but a few. In response to the above findings, the Zambian government has implemented various initiatives to address fertility and related behaviours. The National Population Policy (NPP), first developed in 1989 and subsequently revised in 2007 and 2019, aims to manage population growth. Additionally, the government has invested in family planning (FP) since 1970, as evidenced by the FP 2030 initiative, which focuses on improving access to contraception and other FP services [14].

However, despite these efforts, the Zambia Demographic and Health Survey (ZDHS) of 2018 reveals concerning trends and patterns: 50% of women marry before the age of 19, and 29% of the teenagers are already mothers or expecting their first child. Only 48% of married women use modern contraception, and 20% have unmet need for contraception, in spite of the overall demand of 69% among married women [15]. Further, the abortion rate stands at 45%, with unintended pregnancies ending in abortion increasing from 19% in 1992 to 28% in 2018 [16].

Moreover, the country continues to experience stalled fertility. Previous research has primarily focused on biological and socio-economic factors [11,17,18] to explain this phenomenon, neglecting the intermediate variables proposed by Davis and Blake [19]. Besides, the application of the Bongaarts framework to quantify current fertility outcome (TFR) in these studies makes the use of the Davis and Blake theory more suitable in understanding fertility behaviour as measured by CEB as its more of a cumulative measure [11,13,20]. The Davis and Blake intermediate variables are hypothesized to mediate the relationship between background factors and fertility behaviour. Specifically, they argue that factors such as intercourse (age at first marriage), conception (modern contraceptive use) and gestation (abortion) can significantly influence fertility behaviour among married Zambian women.

By examining these direct determinants, we can better understand why early marriage leads to increased exposure to childbearing [9,12]. Additionally, the use and non-use of modern contraceptives can impact the effectiveness of sexual and reproductive health services, including family planning, ultimately affecting the number of children born [15]. Moreover, access to safe abortion services can play a crucial role in limiting the number of CEB [16,17].

However, these factors are further influenced by social, cultural, and economic factors. For example, limited education and economic dependence can lead to early marriage and increased exposure to childbearing. Furthermore, these factors can hinder women's understanding of family planning and reproductive health, as well as their access and utilisation of modern contraceptive methods and safe abortion services.

While bio-demographic and socio-economic factors are known to influence fertility, this study explores the role of intermediate factors in understanding the stalled fertility transition among married women in Zambia. Given their direct impact on CEB, these intermediate variables should be prioritised when studying women's fertility behaviour in Zambia, including the underlying drivers. Therefore, this study aimed examining the influence of intermediate factors on children ever born while controlling for bio-demographic and socio-economic among married women of child bearing age, to gain a more comprehensive understanding of stalled fertility in Zambia. By identifying these additional factors, policymakers can develop more effective interventions to address high fertility.

## Methodology

This study leveraged data from three nationally representative Zambia Demographic Health Surveys (ZDHSs) conducted in 2007, 2013–14, and 2018. The selection of these specific surveys was based on their inclusion of all essential variables pertinent to the study's objectives and analysis. Furthermore, the women recode files from each ZDHS were acquired from the DHS Program website for use in this study. It is important to note that the ZDHS is a cross-sectional survey employing robust scientific sampling methodologies, ensuring the generalizability of the findings to the entire Zambian population at national and provincial levels.

Particularly, the Demographic and Health Surveys (DHSs) utilize a two-stage cluster sampling design. In the first stage, enumeration areas (EAs) are chosen with probability proportional to size (PPS) using a sampling frame derived from a population census. Subsequently, in the second stage, a predetermined number of households, ranging from 25 to 30, are selected from each chosen EA. Finally, within each sampled household, all eligible women between the ages of 15 and 49 are interviewed upon obtaining informed consent. Furthermore, the research focused exclusively on married women who were both sexually active and fecund. The total weighted sample sizes for each survey year were: 3,533 in 2007, 8,209 in 2013–14, and 6,457 in 2018. As the data were sourced for a single country, there was a possibility of participant overlap across the three surveys. This precluded the researchers from combining the samples for analysis.

### Variables

**Outcome variable.** In this study, "children ever born" was chosen as the outcome to assess married women's fertility behavior. The United Nations defines this term as the average number of live births a woman has had, also known as "lifetime fertility" [21]. During data collection, researchers specifically asked women about the total number of children they had given birth to alive, regardless of whether the children were alive or dead at the time of the survey. The DHS records "children ever born" as a discrete count variable.

**Intermediate variables.** This study made use of the intermediate variables proposed by Davis and Blake in 1956 in their fertility framework and similar studies that have investigated specific intermediate variables within this framework [19,22,23]. Three intermediate variables were chosen for analysis: Age at first marriage, modern contraceptive use, and abortion. These variables were operationalized as follows: Age first marriage "0" less than 18 years and "1" 18 years or older; contraception use "0" No and "1" Using; and Abortion "0" No and "1" Yes.

**Control variables.** The selection of control variables was done based on existing literature conducted in Zambia and other Sub-Saharan African countries [24,25]. These control variables were split into bio-demographic and socio-economic variables.

*Bio-demographic variables:* This study incorporated an examination of the following bio-demographic characteristics of the married women: age of the woman (15–19, 20–24, 25–29, 30–34, 35–39, 40–44 and 45–49); age at first sexual intercourse (less than 18 years and 18 years or older); age at first birth (less than 18 years and 18 years or older); fertility preference (have another, undecided and have nor more); ideal number of children (none, 4 or less and 5 or more); and sex preference (no preference, girl child and boy child).

*Socio-economic variables:* For selection of variables included under this category, this study draws upon established socio-economic variables identified as relevant in prior comparable studies [26–28]. The selected variables encompass: exposure to family planning services (categorized as not exposed and exposed); religious affiliation (Catholic or other, i.e., protestant, Muslim, etc.); educational achievement (no education, primary, secondary, and higher); wealth status (categorized as poor, middle-income, and rich); place of residence (urban or rural); province (Central, Copperbelt, Eastern, Luapula, Lusaka, Muchinga, Northern, North-Western, Southern, and Western). and decision-making autonomy (independent or joint with another individual). Further, it is equally important to acknowledge that both economic empowerment and decision-making autonomy were established as composite variables within the study. These composite variables were constructed by drawing upon sets of questions employed in the DHS, a method consistent with similar research [29].

Specifically, economic empowerment was operationalized based on four DHS questionnaire items: current employment status, respondent earnings, earnings compared to their partner (i.e., earning more), and any level of formal education. Respondents who answered affirmatively to all four questions were classified as fully empowered. Those who responded positively to at least one but not all four questions were categorized as partially empowered. Finally, individuals who answered no to all four questions were coded as not empowered. Following a similar approach, women's decision-making autonomy was constructed using a separate set of four DHS questions. These questions pertained to who held final decision-making authority in the following domains: a woman's healthcare, major household purchases, daily household purchases, and visits to family or relatives. Women who responded affirmatively to all four questions were coded as having decision-making autonomy, while those who responded negatively to any question were coded as lacking autonomy.

## Statistical analysis

This study employed a three-part approach to analyse the data. The first stage involved a descriptive analysis of the married women's background characteristics. Univariate analysis was conducted, generating frequency distributions for each variable across the three survey years. Additionally, means were calculated to establish the number of children ever born (CEB) for women in each survey period. The second stage focused on bivariate analysis. Analysis of Variance (ANOVA) was utilized to identify potential associations between the bio-demographic, socio-economic, and intermediate variables, and the fertility behavior as measured by the number of CEB among married women in Zambia.

In order to determine which among the intermediate, bio-demographic and socio-economic variables held the strongest explanatory power for the observed number of CEB, Negative Binomial Regression (NBR) analysis was employed since CEB is a count discrete variable to produce Adjusted Relative Risk Ratios (AIRRs) associated with each of the aforementioned variables. This particular analytical approach was chosen due to the violation of core assumptions in Ordinary Least Squares (OLS) by count outcomes, such as Children

Ever Born (CEB). Similarly, Poisson regression, though theoretically ideal for this analysis, was equally discarded. During the assessment of its suitability, it was observed that the mean number of children across survey years was consistently lower than the observed variance, rendering the model inappropriate. Under these conditions, the parameter estimates would have exhibited overdispersion. This, in turn, would have led to an underestimation of the standard errors and an overestimation of the test statistics. These preliminary tests violated the assumptions under which the Poisson regression would have been useful had the mean equalled the variance.

Therefore, due to the limitations inherent in Poisson regression, we opted to employ negative binomial regression (NBR) as an alternative modelling approach. NBR explicitly accounts for scenarios involving a series of Bernoulli trials, each resulting in either success or failure. This framework inherently guarantees a predetermined number of successes preceding failures thereby converging to a Poisson distribution if the failures are large. Furthermore, NBR alleviates a restrictive assumption of Poisson regression by permitting the variance of the outcome variable (CEB) to exceed the mean. This relaxation of the assumption proved to be critical for achieving an adequate model fit within the context of this study. The analysis was restricted to models that demonstrated a statistically better fit for each survey year, based on lower observed log-likelihood values, with all dispersions modelled as constants.

Moreover, all the data analysis in this study was done using Stata version 14.2. Prior to the analysis, a multicollinearity test using variance inflation factor (VIF) was performed and it showed that no significant collinearity existed among the independent variables (VIF mean: 1.38, range: 1.01–2.45 in 2007; VIF mean: 1.33, range 1.01–2.13 in 2013–14; and VIF mean: 1.38, range 1.01–2.35 in 2018). The findings are presented as adjusted incidence rate ratio (AIRR) along with corresponding 95% confidence intervals (CIs).

### Ethics approval and consent to participate

This study utilised publicly accessible data sourced from the DHS program's online repository. Ethical clearance for the study was conferred by an Institutional Review Board (IRB) situated in the United States and by local Ethical Review Board in (Tropical Diseases Research Centre) in Ndola Zambia. Voluntary informed consent was gotten from all female participants aged 18 years and older. In the case of participants aged 15 to 17, consent was obtained from both the minor and their parent or guardian.

## Results

The majority of married women in Zambia, across the three survey years, were aged between 20 and 34 years old (Table 1). A significant proportion of these women experience early sexual initiation, with 73.0% reporting initiating sexual activity before the age of 18 in 2018, a decrease from 82.0% in 2007. Moreover, nearly 40% of these women had already become mothers by the time they turned 18. Close to sixty percent of the marred women indicated preference to have another child, whereas more than half of the married women in all the three survey years indicated that their ideal number of children was 5 or more children. Interestingly, there has been a notable decline in the proportion of married women exposed to family planning messages. In 2018, only 23.3% of married women reported exposure to such messages, a significant decrease from 43.7% in 2007.

The proportion of married women with secondary or higher education increased from 25.7% in 2007 to 39.7% in 2018. Besides, across the three survey years, approximately 40% of respondents belonged to either poor or rich households. The majority (60%) of respondents

**Table 1. Percentage distribution of married women aged (15–49 years) by bio-demographic, socio-economic, and intermediate variables (ZDHS: 2007–2018).**

| Variable | 2007 | 2013–14 | 2018 |
|---|---|---|---|
| | N = 3,533 | N = 8,209 | N = 6,457 |
| | Percent | Percent | Percent |
| *Age group* | | | |
| 15–19 | 5.2 | 5.7 | 5.1 |
| 20–24 | 19.9 | 16.7 | 17.9 |
| 25–29 | 24.4 | 22.4 | 20.5 |
| 30–34 | 19.0 | 20.2 | 18.8 |
| 35–39 | 13.5 | 16.4 | 17.4 |
| 40–44 | 9.9 | 11.4 | 12.5 |
| 45–49 | 8.1 | 7.3 | 8.0 |
| *Age at first sexual intercourse* | | | |
| Less than 18 years | 82.0 | 80.0 | 73.0 |
| 18 years or more | 18.0 | 20.0 | 27.0 |
| *Age first birth* | | | |
| Less than 18 years | 38.3 | 39.0 | 38.6 |
| 18 years or more | 61.7 | 61.0 | 61.4 |
| *Fertility preference* | | | |
| Have another | 58.0 | 58.7 | 56.7 |
| Undecided | 6.2 | 4.5 | 5.0 |
| No more | 35.8 | 36.8 | 38.3 |
| *Ideal number of children* | | | |
| None | 7.9 | 4.6 | 3.2 |
| 4 or less | 40.0 | 40.8 | 43.8 |
| 5 or more | 52.1 | 54.5 | 53.0 |
| *Sex preference* | | | |
| No preference | 66.6 | 63.9 | 63.2 |
| Girl child | 19.5 | 21.3 | 21.4 |
| Boy child | 14.0 | 14.8 | 15.3 |
| *Exposure to FP messages* | | | |
| Not expose | 56.3 | 62.9 | 76.7 |
| Exposed | 43.7 | 37.1 | 23.3 |
| *Religion* | | | |
| Catholic | 19.5 | 17.3 | 16.3 |
| Others | 80.5 | 82.7 | 83.7 |
| *Educational level* | | | |
| None | 13.2 | 10.6 | 9.4 |
| Primary | 61.0 | 55.5 | 50.9 |
| Secondary | 21.6 | 29.4 | 34.7 |
| Higher | 4.1 | 4.5 | 5.0 |
| *Wealth index* | | | |
| Poor | 40.6 | 39.4 | 39.3 |
| Middle | 19.9 | 20.1 | 19.3 |
| Rich | 39.4 | 40.4 | 41.4 |
| *Place of residence* | | | |
| Urban | 35.1 | 40.2 | 41.1 |
| Rural | 64.9 | 59.8 | 58.9 |

*(Continued)*

**Table 1.** (Continued)

| Variable | 2007 | 2013–14 | 2018 |
|---|---|---|---|
| | N = 3,533 | N = 8,209 | N = 6,457 |
| | Percent | Percent | Percent |
| *Province* | | | |
| Central | 10.3 | 9.1 | 8.7 |
| Copperbelt | 15.8 | 14.6 | 14.0 |
| Eastern | 16.0 | 13.3 | 13.7 |
| Luapula | 7.9 | 7.3 | 7.9 |
| Lusaka | 14.1 | 18.2 | 18.5 |
| Muchinga | | 5.4 | 5.0 |
| Northern | 14.7 | 5.8 | 6.2 |
| Northwestern | 4.9 | 8.3 | 8.9 |
| Southern | 10.3 | 3.9 | 4.5 |
| Western | 5.9 | 14.0 | 12.6 |
| *Economic empowerment* | | | |
| Not empowered | 6.2 | 4.6 | 4.6 |
| Partially empowered | 84.8 | 86.1 | 84.6 |
| Fully empowered | 9.0 | 9.3 | 10.9 |
| *Decision-making autonomy* | | | |
| Alone or jointly | 36.6 | 53.2 | 56.7 |
| Someone else | 63.4 | 46.8 | 43.3 |
| *Age at first marriage* | | | |
| Less 18 years | 53.7 | 50.5 | 46.7 |
| 18 years or more | 46.3 | 49.5 | 53.3 |
| *Modern contraceptive use* | | | |
| No | 62.4 | 49.3 | 46.1 |
| Yes | 37.6 | 50.7 | 53.9 |
| *Abortion* | | | |
| No | 83.0 | 86.9 | 88.1 |
| Yes | 17.0 | 13.1 | 11.9 |

resided in rural areas. Lusaka and Copperbelt provinces consistently had higher proportion of married women compared to Luapula and North-Western provinces. Moreover, the percentage of economically empowered married women slightly increased from 9.0% in 2007 to 10.9% in 2018. Furthermore, the proportion of married women with sole or joint decision-making autonomy rose from 36.6% in 2007 to 56.7% in 2018.

With regard to the intermediate variables, the proportion of women marrying before the age of 18 decreased from 53.7% in 2007 to 46.7% in 2018. Concurrently, there was a substantial increase modern contraceptive use among married women, rising from 37.6% in 2007 to over half (53.9%) in 2018. Further, the proportion of married women a reporting prior abortion reduced from 17.0% to 11.9% over the same period.

## Mean number of children ever born in Zambia

Fig 1 below shows the average number of children a woman was expected to have across three survey years (2007–2018). The results reveal that, on average, the number CEB was approximately four. Specifically, the figure shows that there was a slight decrease in the average CEB

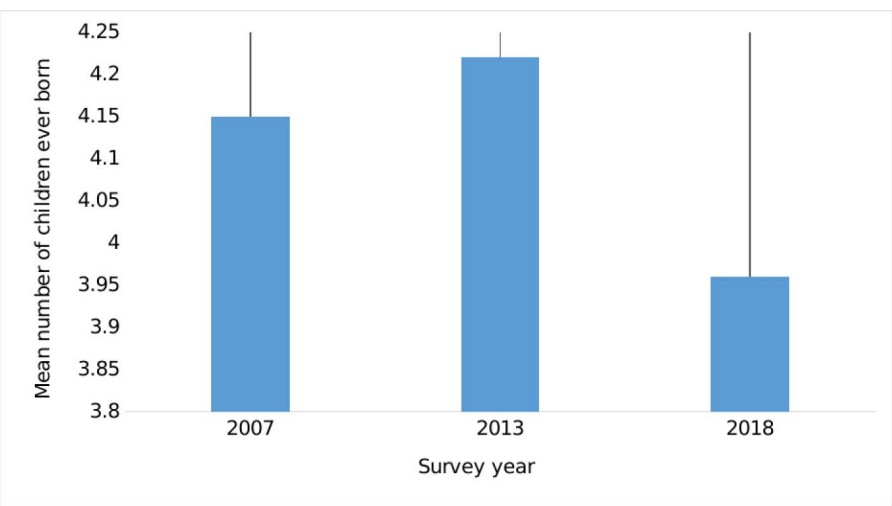

**Fig 1. Mean number of children ever born with corresponding standard deviations among married women (ZDHS: 2007–2018).**

between 2007 (4.2 ± 2.7) and 2018 (4.0 ± 2.5), suggesting a trend towards fewer children over this period.

**Bio-demographic, socio-economic and intermediate correlates of children ever born.** The mean number of children ever born among married women in Zambia increased with age, with those aged 40 years and above having an average of 7 children, with a standard deviation of 3, across the survey years. Apart from age, various bio-demographic, socio-economic and intermediate factors were correlated with the number of CEB among married women. A bivariate analysis revealed that all variables, except for sex preference and economic empowerment, were statistically significant (Table 2).

Specifically, this study found a statistically significant difference in the mean number of CEB between women who initiated sexual activity or childbirth before 18 years of age (4.3 and 5.0 in 2007; 4.3 and 4.7 in 2018) and those who did so at 18 years or older (3.5 and 3.6 in 2007; 3.4 and 3.6 in 2018), p < 0.001. An examination of fertility preferences reveals that women desiring no further childbearing had a mean number CEB twice that of women seeking an additional child in both 2007 (3.0 compared to 6.1) and 2018 (2.7 compared to 5.8). Similarly, married women who ideally wanted no children had a significantly higher mean CEB than those who ideally wanted five or more children in both 2007 (5.1 vs 4.9) and 2018 (5.3 and 4.9), p < 0.001. Additionally, the study examined the impact of exposure to family planning (FP) messages on CEB. Women exposed to FP messages in 2013–14 and 2018 had a statistically significant lower average number of CEB compared to the unexposed group (4.4 vs 4.0 and 4.1 vs 3.9, respectively) (p < 0.001). However, this trend was not observed in 2007.

An inverse association was observed between educational attainment and average number of children ever born (CEB) among married women. Women with secondary and higher education reported having an average of two or three children, respectively, compared to women with no education or primary education, who averaged five children across all survey years (p < 0.001). Married women from rich households reported having, on average, one less child compared to those from poorer households (4 vs. 5 in 2007; 3 vs. 4 in 2018). Likewise, urban women had slightly fewer number of CEB than rural women. In 2013–14 and 2018, married women in Lusaka, Copperbelt, and Muchinga provinces had fewer number of CEB in

**Table 2.** Mean number of children ever born to ever married women aged 15–49 and women aged 40–49 by bio-demographic, socio-economic, and cultural characteristics (ZDHS: 2007–2018).

| Variable | 2007 | | | 2013–14 | | | 2018 | | |
|---|---|---|---|---|---|---|---|---|---|
| | Mean CEB | SD | F-value | Mean CEB | SD | F-value | Mean CEB | SD | F-value |
| **Bio-demographic variables** | | | | | | | | | |
| *Age group* | | | | | | | | | |
| 15–19 | 1.0 | 0.7 | 579.9*** | 1.1 | 0.6 | 1336.9*** | 1.1 | 0.6 | 1075.9*** |
| 20–24 | 2.0 | 1.0 | | 2.0 | 1.0 | | 1.8 | 0.9 | |
| 25–29 | 3.2 | 1.4 | | 3.2 | 1.4 | | 3.0 | 1.3 | |
| 30–34 | 4.5 | 1.9 | | 4.5 | 1.8 | | 4.2 | 1.7 | |
| 35–39 | 5.9 | 2.3 | | 4.7 | 2.3 | | 5.3 | 2.1 | |
| 40–44 | 6.9 | 2.8 | | 6.7 | 2.7 | | 6.2 | 2.6 | |
| 45–49 | 7.2 | 3.0 | | 7.4 | 2.7 | | 7.1 | 2.6 | |
| *Age at first sexual intercourse* | | | | | | | | | |
| Less than 18 years | 4.3 | 2.7 | 48.4*** | 4.4 | 2.7 | 177.9*** | 4.3 | 2.6 | 174.2*** |
| 18 years or more | 3.5 | 2.3 | | 3.5 | 2.3 | | 3.4 | 2.3 | |
| *Age first birth* | | | | | | | | | |
| Less than 18 years | 5.0 | 2.7 | 231.4*** | 4.9 | 2.7 | 342.4*** | 4.7 | 2.7 | 257.9*** |
| 18 years or more | 3.6 | 2.4 | | 3.8 | 2.5 | | 3.6 | 2.4 | |
| *Fertility preference* | | | | | | | | | |
| Have another | 3.0 | 1.9 | 744.1*** | 3.0 | 1.9 | 2017.2*** | 2.7 | 1.8 | 1675.3*** |
| Undecided | 4.2 | 2.4 | | 4.8 | 2.2 | | 4.8 | 2.3 | |
| No more | 6.1 | 2.6 | | 6.2 | 2.5 | | 5.8 | 2.4 | |
| *Ideal number of children* | | | | | | | | | |
| None | 5.1 | 2.9 | 238.3*** | 6.0 | 3.0 | 915.3*** | 5.3 | 3.0 | 654.9*** |
| 4 or less | 3.0 | 2.2 | | 2.9 | 2.0 | | 2.8 | 1.9 | |
| 5 or more | 4.9 | 2.6 | | 5.1 | 2.5 | | 4.9 | 2.5 | |
| *Sex preference* | | | | | | | | | |
| No preference | 4.2 | 2.7 | 4.6$^{NS}$ | 4.3 | 2.7 | 0.5$^{NS}$ | 4.1 | 2.6 | 5.2$^{NS}$ |
| Girl child | 3.9 | 2.6 | | 4.2 | 2.6 | | 3.9 | 2.5 | |
| Boy child | 4.1 | 2.6 | | 4.2 | 2.6 | | 4.0 | 2.5 | |
| **Socio-economic variables** | | | | | | | | | |
| *Exposure to FP messages* | | | | | | | | | |
| Not expose | 4.2 | 2.7 | 5.3$^{NS}$ | 4.4 | 2.7 | 38.5*** | 4.1 | 2.6 | 10.9*** |
| Exposed | 4.0 | 2.6 | | 4.0 | 2.5 | | 3.9 | 2.4 | |
| *Religion* | | | | | | | | | |
| Catholic | 4.3 | 2.8 | 2.7** | 4.4 | 2.7 | 2.9$^{NS}$ | 4.0 | 2.6 | 1.1$^{NS}$ |
| Others | 4.1 | 2.6 | | 4.2 | 2.6 | | 4.1 | 2.5 | |
| *Educational level* | | | | | | | | | |
| None | 4.9 | 2.8 | 79.8*** | 5.4 | 2.8 | 372.9*** | 5.3 | 2.7 | 318.2*** |
| Primary | 4.5 | 2.7 | | 4.8 | 2.6 | | 4.6 | 2.6 | |
| Secondary | 3.2 | 2.2 | | 3.1 | 2.1 | | 3.0 | 1.9 | |
| Higher | 2.4 | 1.6 | | 2.4 | 1.5 | | 2.5 | 1.4 | |
| *Wealth index* | | | | | | | | | |
| Poor | 4.6 | 2.7 | 41.1** | 4.7 | 2.7 | 172.4*** | 4.4 | 2.7 | 142.8*** |
| Middle | 4.3 | 2.7 | | 4.7 | 2.7 | | 4.4 | 2.7 | |
| Rich | 3.7 | 2.5 | | 3.6 | 2.3 | | 3.3 | 2.1 | |

*(Continued)*

**Table 2.** (Continued)

| Variable | 2007 | | | 2013–14 | | | 2018 | | |
|---|---|---|---|---|---|---|---|---|---|
| | Mean CEB | SD | F-value | Mean CEB | SD | F-value | Mean CEB | SD | F-value |
| *Place of residence* | | | | | | | | | |
| Urban | 3.7 | 2.5 | 56.0* | 3.6 | 2.3 | 354.8*** | 3.4 | 2.1 | 255.6*** |
| Rural | 4.4 | 2.7 | | 4.7 | 2.7 | | 4.4 | 2.7 | |
| *Province* | | | | | | | | | |
| Central | 4.0 | 2.7 | 3.8^NS | 4.5 | 2.8 | 18.1*** | 4.1 | 2.5 | 15.9*** |
| Copperbelt | 4.2 | 2.6 | | 4.0 | 2.4 | | 3.7 | 2.4 | |
| Eastern | 4.3 | 2.6 | | 4.2 | 2.6 | | 3.9 | 2.5 | |
| Luapula | 4.5 | 2.8 | | 4.6 | 2.6 | | 4.6 | 2.8 | |
| Lusaka | 3.6 | 2.5 | | 3.5 | 2.2 | | 3.4 | 2.1 | |
| Muchinga | – | – | | 4.0 | 2.6 | | 4.0 | 2.5 | |
| Northern | 4.4 | 2.8 | | 4.5 | 2.8 | | 4.5 | 2.8 | |
| Northwestern | 4.3 | 2.7 | | 4.8 | 2.8 | | 4.4 | 2.6 | |
| Southern | 4.0 | 2.6 | | 4.3 | 2.6 | | 4.0 | 2.4 | |
| Western | 4.0 | 2.5 | | 4.2 | 2.6 | | 4.1 | 2.6 | |
| *Economic empowerment* | | | | | | | | | |
| Not empowered | 4.8 | 2.7 | 6.3^NS | 5.2 | 2.7 | 32.1^NS | 5.1 | 2.7 | 36.3^NS |
| Partially empowered | 4.1 | 2.6 | | 4.2 | 2.6 | | 3.9 | 2.5 | |
| Fully empowered | 4.1 | 2.6 | | 4.6 | 2.7 | | 4.3 | 2.6 | |
| *Decision-making autonomy* | | | | | | | | | |
| Alone or jointly | 4.1 | 2.6 | 0.1^NS | 4.2 | 2.6 | 2.3* | 4.0 | 2.5 | 1.4*** |
| Someone else | 4.2 | 2.7 | | 4.3 | 2.7 | | 4.1 | 2.6 | |
| **Intermediate variables** | | | | | | | | | |
| *Age at first marriage* | | | | | | | | | |
| Less 18 years | 4.7 | 2.8 | 166.1*** | 4.8 | 2.7 | 417.7*** | 4.6 | 2.7 | 273.9*** |
| 18 years or more | 3.5 | 2.4 | | 3.7 | 2.4 | | 3.6 | 2.3 | |
| *Modern contraceptive use* | | | | | | | | | |
| No | 4.2 | 2.8 | 4.6*** | 4.4 | 2.8 | 22.7*** | 4.1 | 2.8 | 8.4*** |
| Yes | 4.0 | 2.4 | | 4.1 | 2.4 | | 4.0 | 2.3 | |
| *Abortion* | | | | | | | | | |
| No | 4.0 | 2.6 | 28.8* | 4.2 | 2.6 | 44.6*** | 4.0 | 2.5 | 5.9^NS |
| Yes | 4.7 | 2.8 | | 4.7 | 2.9 | | 4.2 | 2.6 | |

SD standard deviation; ***p < 0.001; **p < 0.01; *p < 0.05; NS = non-significant and – no response for that category.

than to those in North-Western, Luapula, and Northern Provinces (p < 0.001). Furthermore, decision-making power within the household appeared to play a role. Married women who reported making decisions alone or jointly with their husbands tended to have slightly fewer CEB than those who did not (except in 2007).

An analysis of intermediate variables and the number of CEB among married women revealed significant associations. Early marriage, defined as marrying before the age of 18, was linked to a substantially higher mean number of CEB (five) across all three survey years. This was in stark contrast to women who married at 18 years or older, who had an average of four CEB (p < 0.001). Furthermore, the study demonstrated a statistically significant (p < 0.001) association between modern contraceptive use and lower fertility rates. Women who reported using modern contraception at the time of the survey had a slightly lower average number of

CEB compared to those who did not. Contrarily, a negative correlation between a history of abortion and the average number of CEB. Married women who disclosed having had a prior abortion exhibited a higher average number of CEB (4.7 in both 2007 and 2013–14) compared to those who did not (4.0 in 2007 and 4.2 in 2013).

## Factors associated with children ever born

Model I examine the relationship between the intermediate variables and the number of children ever born among married women in Zambia across the three survey years.

Data from three separate surveys were analysed and the findings indicate a statistically significant association between early marriage and increased CEB. Married women who married before the age of 18 consistently had a higher likelihood of having more CEB compared to those who married at 18 or older. This trend is consistent across all three survey years, as reflected in the consistently elevated adjusted incidence risk ratios (AIRRs) for the early marriage group. For instance, the AIRR in 2007 was 1.30 (with a 95% confidence interval of 1.24 to 1.36), remaining similarly high in 2013–14 and 2018 (1.24, 95% CI: 1.24–1.33). Conversely, the use of modern contraception methods was not statistically significantly associated with CEB, except in 2018. In that year, married women who were not using modern method of contraception had a higher likelihood of having more CEB than those who used no method (AIRR = 1.04, 95% CI: 1.01–1.08). Furthermore, our analysis revealed a statistically significant association between a history of abortion and increased number of CEB. However, this elevated risk appeared to diminish over time. While the AIRR for women with a history of abortion was the course of the three survey periods 1.15 (95% CI: 1.08–1.23) in 2007 it deceased to 1.12 (95% CI: 1.06–1.17) in 2013–14; and 1.07 (95% CI: 1.01–1.13) in 2018.

In Model II, which accounts for the influence of bio-demographic factors, extends the analysis beyond the relationship between intermediate factors and the number of CEB to women in Zamba. Interestingly, the model reveals an unexpected association between modern contraception use and CEB. Contrary to expectations, married women who consistently did not use modern contraception across the study period (2007, 2013, and 2018) were less likely to have more children compared to those who did use contraception (AIRR = 0.92, 95% CI: 0.89–0.95; AIRR = 0.98, 95% CI: 0.96–1.00; and AIRR = 0.96, 95% CI: 0.94–0.98, respectively). Similarly, married women who reported having had a prior abortion exhibited a lower risk of having more CEB in 2018 compared to those without a history of abortion (AIRR = 0.95,

**Model I. Adjusted incidence rate ratios of intermediate variables associated with children ever born among married women in Zambia, ZDHS (2007–2018).**

| Variables | 2007 | | 2013–14 | | 2018 | |
|---|---|---|---|---|---|---|
| | AIRR | 95% CI | AIRR | 95% CI | AIRRs | 95% CI |
| **Intermediate variables** | | | | | | |
| *Age at first marriage* | | | | | | |
| Less than 18 years | 1.30*** | 1.24–1.36 | 1.28*** | 1.24–1.33 | 1.28*** | 1.24–1.33 |
| 18 years or more | 1 | | 1 | | 1 | |
| *Modern contraception use* | | | | | | |
| No | 1.00 | 0.96–1.05 | 1.03 | 1.00–1.07 | 1.04* | 1.01–1.08 |
| Yes | 1 | | 1 | | 1 | |
| *Ever had an abortion* | | | | | | |
| No | 1 | | 1 | | 1 | |
| Yes | 1.15*** | 1.08–1.23 | 1.12*** | 1.06–1.17 | 1.07* | 1.01–1.13 |

***p < 0.001 and *p < 0.05; AIRR = adjusted incidence risk ratio; and CI = 95% confidence interval.

**Model II. Adjusted incidence rate ratios of the intermediate and bio-demographic variables associated with married women fertility behaviour as measured by children ever born in Zambia, ZDHS (2007–2018).**

| Variables | 2007 | | 2013–14 | | 2018 | |
|---|---|---|---|---|---|---|
| | AIRRs | CI | AIRRs | CI | AIRRs | CI |
| **Intermediate variables** | | | | | | |
| *Age at first marriage* | | | | | | |
| Less than 18 years | 1.14*** | 1.10–1.18 | 1.14*** | 1.11–1.17 | 1.12*** | 1.09–1.15 |
| 18 years or more | 1 | | 1 | | 1 | |
| *Contraception use* | | | | | | |
| No | 0.92*** | 0.89–0.95 | 0.98* | 0.96–1.00 | 0.96*** | 0.94–0.98 |
| Yes | 1 | | 1 | | 1 | |
| *Ever had an abortion* | | | | | | |
| No | 1 | | 1 | | 1 | |
| Yes | 0.97 | 0.93–1.01 | 0.98 | 0.95–1.01 | 0.95** | 0.92–0.98 |
| **Bio-demographic variables** | | | | | | |
| *Age of respondent* | | | | | | |
| 15–19 | 1 | | 1 | | 1 | |
| 20–24 | 2.02*** | 1.81–2.24 | 1.94*** | 1.82–2.06 | 1.83*** | 1.71–1.96 |
| 25–29 | 3.09*** | 2.78–3.44 | 2.94*** | 2.76–3.13 | 2.80*** | 2.61–3.00 |
| 30–34 | 3.99*** | 3.58–4.45 | 3.75*** | 3.51–4.01 | 3.65*** | 3.40–3.92 |
| 35–39 | 4.79*** | 4.26–5.38 | 4.41*** | 4.09–4.75 | 4.22*** | 3.92–4.55 |
| 40–44 | 5.36*** | 4.75–6.05 | 4.90*** | 4.55–5.28 | 4.65*** | 4.29–5.04 |
| 45–49 | 5.30*** | 4.69–5.99 | 5.11*** | 4.72–5.54 | 5.11*** | 4.70–5.55 |
| *Age at first sexual intercourse* | | | | | | |
| Less than 18 years | 1.06* | 1.01–1.10 | 1.07*** | 1.04–1.11 | 1.09*** | 1.06–1.13 |
| 18 years or more | 1 | | 1 | | 1 | |
| *Age at first birth* | | | | | | |
| Less than 18 years | 1.24*** | 1.20–1.28 | 1.17*** | 1.14–1.20 | 1.18*** | 1.15–1.21 |
| 18 years or more | 1 | | 1 | | 1 | |
| *Fertility preference* | | | | | | |
| Have another | 1 | | 1 | | 1 | |
| Undecided | 1.14*** | 1.07–1.21 | 1.18*** | 1.13–1.23 | 1.26*** | 1.21–1.32 |
| No more | 1.31*** | 1.26–1.36 | 1.29*** | 1.26–1.33 | 1.32*** | 1.28–1.36 |
| *Ideal number of children* | | | | | | |
| None | 1.30*** | 1.22–1.38 | 1.40*** | 1.34–1.47 | 1.37*** | 1.29–1.45 |
| 4 or less | 1 | | 1 | | 1 | |
| 5 or more | 1.32*** | 1.27–1.36 | 1.37*** | 1.34–1.41 | 1.36*** | 1.32–1.39 |
| *Sex preference* | | | | | | |
| No preference | 1 | | 1 | | 1 | |
| Girl child | 0.94** | 0.91–0.98 | 0.95*** | 0.93–0.97 | 0.94*** | 0.91–0.96 |
| Boy child | 0.95* | 0.92–0.99 | 0.95*** | 0.92–0.97 | 0.94*** | 0.92–0.97 |

***p < 0.001; **p < 0.01; and *p < 0.05; AIRR = adjusted incidence risk ratio; and CI = 95% confidence interval.

95% CI: 0.92–0.98). However, one exception to the results observed in this model is age at first marriage, which was not found to be significantly related to CEB in this model.

The analysis of married women's age and the number of CEB across three survey years reveals a positive correlation. Women aged 40 and above consistently exhibited the highest

number of CEB (5) across all survey years (p < 0.001). Furthermore, the data further shows that women who began childbearing before the age of 18 were significantly more likely to have a higher number of CEB compared to those who delayed childbearing until 18 or later. This trend persisted throughout the surveys, with statistically significant AIRR exceeding 1.00 in all years (p < 0.001). Similarly, women whose fertility preference was undecided or have no more children; or that their ideal number of children was none or five or more, exhibited higher likelihood of having a higher number of CEB compared to those who indicated wanting another or four or fewer (p < 0.001).

Model III delves deeper into the analysis by integrating intermediate, bio-demographic and socio-economic variables to comprehensively assess their relationship with the number of CEB to married women in Zambia across the three survey years.

This model aligns with the findings of Model II, which revealed that married women who entered matrimony at or before the age of 18 were significantly more likely to have a higher number of children compared to those who married after 18 across all survey years (AIRR = 1.10, 95% CI: 1.07–1.14 in 2007; AIRR = 1.10, 95% CI: 1.10–1.12 in 2013; and AIRR = 1.07, 95% CI: 1.05–1.10). On the contrary, our findings contradict the expected association between modern contraception use among married women and their total number of CEB. This pattern persisted across all three survey years (2007, 2013, and 2018). Notably, women who reported not using modern contraception were consistently less likely to have a larger number of CEB compared to those who did use modern contraception (AIRR = 0.88, 95% CI: 0.85–0.91 in 2007; AIRR = 0.94, 95% CI: 0.93–0.96 in 2013; and AIRR = 0.94, 95% CI: 0.92–0.96 in 2018, respectively). Similarly, a history of abortion among married women did not show a statistically significant association with the total number of CEB, except in 2018. In that year, married women who reported having had a prior abortion were more likely to have fewer CEB compared to those who did not report a prior abortion (AIRR = 0.94, 95% CI: 0.91–0.97).

Further, the study shows that various bio-demographic variables were associated with higher incidence of CEB among married women in Zambia. The likelihood of married women having a higher number of CEB increased with age, with women aged 40 years and above having higher incidence rate than those aged 15–24 years across all the survey years (p < 0.001). Contrary to what was observed in Model II, a statistically insignificant association was observed between women age at first sex and number of CEB, except for 2018 were women who initiated sex before turning 18 years had a higher likelihood of having more CEB than those who began at 18 years and above (AIRR = 1.04, 95% CI: 1.01–1.07). Further, married who begun child bearing before 18 years consistently exhibited higher numbers of CEB than those who gave birth at 18 years or above across all the survey years (AIRR = 1.23, 95% CI: 1.19–1.27 in 2007; AIRR = 1.16, 95% CI: 1.14–1.18 in 2013; and AIRR = 1.18, 95% CI: 1.15–1.21). In addition to the aforementioned findings, the model's analysis further suggests that married women whose fertility preference was undecided or no more children are more likely to have a higher number of CEB than those who still intend to have another child. Equally, the study also reveals that married women whose ideal number of children was to have another child or have five or more children were more likely to have a higher number of CEB compared to those who reported that they have ideal number of children was 4 or less. This trend was statistically significant across all the three survey years (p < 0.001). Conversely, married women who expressed a preference for a girl or boy child were less likely to have a higher number of CEB compared to those who had no preference. This association was statistically significant across the three survey years.

Besides the intermediate and bio-demographic factors associated with CEB, the model further reveals a significant association between socio-economic variables and the number of children ever born to married women. These variables included religious affiliation,

**Model III. Adjusted incidence rate ratios of the intermediate, bio-demographic and socio-economic variables associated with married women fertility behaviour as measured by children ever born in Zambia, ZDHS (2007–2018).**

| Variables | 2007 | | 2013 | | 2018 | |
|---|---|---|---|---|---|---|
| | AIRRs | CI | AIRRs | CI | AIRRs | CI |
| **Intermediate variables** | | | | | | |
| *Age at first marriage* | | | | | | |
| Less than 18 years | 1.10*** | 1.07–1.14 | 1.10*** | 1.07–1.12 | 1.07*** | 1.05–1.10 |
| 18 years or more | 1 | | 1 | | 1 | |
| *Modern contraception use* | | | | | | |
| No | 0.88*** | 0.85–0.91 | 0.94*** | 0.93–0.96 | 0.94*** | 0.92–0.96 |
| Yes | 1 | | 1 | | 1 | |
| *Ever had an abortion* | | | | | | |
| No | 1 | | 1 | | 1 | |
| Yes | 0.98 | 0.94–1.02 | 0.97 | 0.95–1.00 | 0.94*** | 0.91–0.97 |
| **Bio-demographic variables** | | | | | | |
| *Age of respondent* | | | | | | |
| 15–19 | 1 | | 1 | | 1 | |
| 20–24 | 2.00*** | 1.80–2.23 | 1.94*** | 1.82–2.06 | 1.89*** | 1.77–2.03 |
| 25–29 | 3.12*** | 2.81–3.47 | 2.99*** | 2.81–3.18 | 3.02*** | 2.82–3.24 |
| 30–34 | 4.03*** | 3.62–4.50 | 3.87*** | 3.63–4.14 | 3.97*** | 3.69–4.26 |
| 35–39 | 4.84*** | 4.31–5.43 | 4.51*** | 4.20–4.84 | 4.62*** | 4.30–4.97 |
| 40–44 | 5.46*** | 4.84–6.17 | 4.98*** | 4.63–5.37 | 5.06*** | 4.68–5.46 |
| 45–49 | 5.44*** | 4.82–6.14 | 5.29*** | 4.90–5.71 | 5.49*** | 5.08–5.94 |
| *Age at first sexual intercourse* | | | | | | |
| Less than 18 years | 1.03 | 0.99–1.07 | 1.01 | 0.98–1.03 | 1.04** | 1.01–1.07 |
| 18 years or more | 1 | | 1 | | 1 | |
| *Age at first birth* | | | | | | |
| Less than 18 years | 1.23*** | 1.19–1.27 | 1.16*** | 1.14–1.18 | 1.18*** | 1.15–1.21 |
| 18 years or more | 1 | | 1 | | 1 | |
| *Fertility preference* | | | | | | |
| Have another | 1 | | 1 | | 1 | |
| Undecided | 1.16*** | 1.09–1.23 | 1.17*** | 1.13–1.23 | 1.23*** | 1.18–1.28 |
| No more | 1.33*** | 1.29–1.38 | 1.29*** | 1.26–1.33 | 1.31*** | 1.28–1.35 |
| *Ideal number of children* | | | | | | |
| None | 1.19*** | 1.12–1.26 | 1.26*** | 1.21–1.32 | 1.24*** | 1.17–1.30 |
| 4 or less | 1 | | 1 | | 1 | |
| 5 or more | 1.22*** | 1.18–1.26 | 1.25*** | 1.23–1.29 | 1.22*** | 1.19–1.25 |
| *Sex preference* | | | | | | |
| No preference | 1 | | 1 | | 1 | |
| Girl child | 0.95** | 0.92–0.98 | 0.97** | 0.94–0.99 | 0.95*** | 0.93–0.98 |
| Boy child | 0.95* | 0.92–0.99 | 0.96** | 0.94–0.99 | 0.95** | 0.93–0.98 |
| **Socio-economic variables** | | | | | | |
| *Exposure to FP messages* | | | | | | |
| Not expose | 1.01 | 0.98–1.04 | 1.01 | 0.99–1.03 | 1.01 | 0.99–1.03 |
| Exposed | 1 | | 1 | | 1 | |
| *Religion* | | | | | | |
| Catholic | 1.00 | 0.97–1.04 | 0.98 | 0.95–1.00 | 0.97* | 0.94–1.00 |
| Others | 1 | | 1 | | 1 | |

*(Continued)*

**Model III.** (Continued)

| Variables | 2007 | | 2013 | | 2018 | |
|---|---|---|---|---|---|---|
| | AIRRs | CI | AIRRs | CI | AIRRs | CI |
| *Educational level* | | | | | | |
| None | 1.50*** | 1.31–1.72 | 1.44*** | 1.34–1.55 | 1.39*** | 1.30–1.49 |
| Primary | 1.46*** | 1.28–1.66 | 1.40*** | 1.31–1.49 | 1.34*** | 1.26–1.42 |
| Secondary | 1.33*** | 1.18–1.50 | 1.24*** | 1.16–1.32 | 1.22*** | 1.16–1.28 |
| Higher | 1 | | 1 | | 1 | |
| *Wealth index* | | | | | | |
| Poor | 1.14*** | 1.08–1.20 | 1.10*** | 1.07–1.14 | 1.11*** | 1.07–1.15 |
| Middle | 1.07** | 1.02–1.12 | 1.09*** | 1.05–1.12 | 1.07*** | 1.03–1.11 |
| Rich | 1 | | 1 | | 1 | |
| *Place of residence* | | | | | | |
| Urban | 1 | | 1 | | 1 | |
| Rural | 1.02 | 0.98–1.07 | 1.08*** | 1.05–1.11 | 1.09*** | 1.06–1.13 |
| *Province* | | | | | | |
| Central | 1.04 | 0.98–1.10 | 1.07** | 1.03–1.12 | 1.03 | 0.98–1.08 |
| Copperbelt | 1.06* | 1.01–1.12 | 1.01 | 0.97–1.05 | 1.03 | 0.98–1.07 |
| Eastern | 0.97 | 0.92–1.02 | 1.02 | 0.98–1.07 | 1.02 | 0.99–1.06 |
| Luapula | 1.11*** | 1.05–1.17 | 1.04 | 0.99–1.09 | 1.10*** | 1.06–1.14 |
| Lusaka | 1 | | 1 | | 1 | |
| Muchinga | – | – | 0.96 | 0.92–1.01 | 1.02 | 0.97–1.06 |
| Northern | 1.05 | 1.00–1.10 | 1.06** | 1.01–1.11 | 1.08*** | 1.03–1.12 |
| Northwestern | 1.05 | 0.99–1.11 | 1.10*** | 1.05–1.14 | 1.08** | 1.03–1.13 |
| Southern | 1.04 | 0.98–1.10 | 1.05* | 1.00–1.10 | 1.07** | 1.02–1.12 |
| Western | 0.95 | 0.89–1.02 | 1.05 | 1.00–1.10 | 1.06** | 1.02–1.11 |
| *Economic empowerment* | | | | | | |
| Not empowered | 0.99 | 0.92–1.07 | 0.94 | 0.88–1.01 | 0.98 | 0.92–1.04 |
| Partially empowered | 1.04 | 0.99–1.10 | 1.00 | 0.97–1.03 | 1.03 | 1.00–1.06 |
| Fully empowered | 1 | | 1 | | 1 | |
| *Decision-making autonomy* | | | | | | |
| Alone or jointly | 1 | | 1 | | 1 | |
| Someone else | 1.00 | 0.97–1.03 | 1.03** | 1.01–1.05 | 1.04*** | 1.02–1.07 |

***p < 0.001; **p < 0.01; *p < 0.05; – no response for that category; AIRR = adjusted incidence risk ratio; and CI = 95% confidence interval.

educational attainment, household wealth index, place of residence, provincial location, and decision-making autonomy. The observed associations were consistent across all three surveys, with some variations present in individual survey years. In particular married Catholic women exhibited a statistically significant lower likelihood of having a higher number of CEB compared to married women affiliated with other religions (AIRR = 0.97, 95% CI: 0.94–1.00). Contrarily, an inverse association was observed for educational attainment. Married women in Zambia with lower levels of education displayed a significantly higher number of children ever born across all three survey years (p < 0.001). Household wealth index also played a role, with married women from poor and middle-class households having a statistically significant higher number of children ever born than their counterparts from rich households (p < 0.001).

In addition, in both 2013 and 2018, married women residing in rural areas exhibited a significantly higher likelihood of having a greater number of children ever born (CEB)

compared to their urban counterparts. This finding was consistent across both years, with an adjusted incidence rate ratio (AIRR = 1.08, 95% CI: 1.05–1.11 in 2013 and AIRR = 1.09, 95%: 1.06–1.13 in 2018) in 2018. The only exception to this pattern was observed in 2007. The relationship between province of residence and number of CEB for married women varied across survey years. In 2007, married women residing in Copperbelt and Luapula provinces exhibited a greater likelihood of having more CEB compared to their counterparts in Lusaka province (AIRR = 1.06, 95% CI: 1.01–1.12 and AIRR = 1.11, 95% CI: 1.05–1.17, respectively). In the year 2013, married women residing in Central, Norther, North-Western, and Southern provinces of Zambia exhibited a statistically significant greater likelihood of having more CEB compared to their counterparts in Lusaka province (AIRR = 1.07, 95% CI: 1.03–1.12; AIRR = 1.06, 95% CI: 1.01–1.11; AIRR = 1.10, 95% CI: 1.05–1.14; and AIRR = 1.05, 95% CI: 1.00–1.10). A similar trend persisted for in 2018, married women from Luapula, Northern, North-Western, Southern and Western provinces displayed a higher likelihood of having more children ever born compared to those in Lusaka province (AIRR = 1.10, 95% CI: 1.06–1.14; AIRR = 1.08, 95% CI: 1.03–1.12; AIRR = 1.08, 95% CI: 1.03–1.13; AIRR = 1.07, 95% CI: 1.02–1.12; and AIRR = 1.06, 95% CI: 1.02–1.11).

Furthermore, a statistically significant association was observed between married women's level of decision-making autonomy and their total number of CEB, with the exception of data from 2007. Specifically, in both 2013 and 2018, married women who reported lacking independent decision-making power (decisions made by someone else) exhibited a higher likelihood of having a greater number of CEB compared to those who reported making decisions solely themselves or jointly with their husbands (AIRR = 1.03, 95% CI: 1.01–1.05 and AIRR = 1.04, 95% CI: 1.02–1.07, respectively).

## Discussion

This study investigated the influence of bio-demographic, socio-economic, intermediate variables on the number of children ever born (CEB) among married women in Zambia. While the average CEB has remained relatively stable at around four children per woman, this figure is intermediate compared to other studies in Sub-Saharan Africa (SSA). For instance, a study in SSA reported a range of 2.9 to 7.8 children per woman, with South Africa and Niger representing the extremes [30]. Therefore, this finding of ours is indicative of the fact that there might be various factors at play determining the number of CEB in Zambia.

Among the intermediate variables considered in this study, early marriage, particularly before the age of 18, is significantly associated with higher completed family size (CEB) in this study. This finding aligns with previous research in Sub-Saharan Africa (SSA) [31,32]. Biological factors among married women, such as peak fertility in their 20s, contribute to this correlation, as early marriage allows for a longer reproductive window [33]. Additionally, social factors can play a role. Societal norms often encourage early marriage and larger families, particularly among poorer women with limited educational opportunities and economic dependence on men [25,34]. These women may feel pressured to marry young and prioritize childbearing, leading to higher CEB [25]. However, it's important to note that correlation does not equal causation. While early marriage is associated with higher CEB, factors like education and employment can influence fertility outcomes. Women who marry young but pursue education and employment may have greater control over their reproductive choices, potentially leading to lower CEB [35].

Our study, while statistically significant, revealed an unexpected relationship between modern contraception use and the total number of children ever born among married Zambian women. This finding contradicts the expected positive association between non-use and higher fertility, a trend observed in similar studies [36]. A plausible explanation for this

discrepancy lies in the primary purpose of modern contraception use. While married women may utilise modern contraceptive methods, their intent may not be for limiting the number of CEB but rather to space births [26]. This suggest that women may still achieve their desired number of children, irrespective of them using the modern contraceptives. Moreover, the unmet need for contraception among women who desire family limitation could contribute to the observed number of children ever born [37].

Furthermore, this study reinforces the established connection between abortion and reduced fertility among married women and aligns with previous studies conducted across both developed and developing nations [38]. Several factors may contribute to this link. Firstly, limited access to and utilisation of family planning resources can lead to unintended pregnancies, resulting in women having more CEB than desired [39,40]. Secondly, cultural and religious beliefs that discourage safe abortion as a means of family planning can further exacerbate this issue, as abortion is abhorred because life is regarded as sacred and devoid of human control [41,42]. Additionally, socio-economic factors, such as poverty and limited education may influence married women's reproductive decisions. In Zambia, for instance, women from disadvantaged backgrounds may value larger families due to the potential economic and social benefits of having more children, particularly in agricultural communities or as a form of old-age security.

Older, married women in Zambia tend to have more children compared to younger women, a pattern observed in other studies [43,44]. This is expected because older women compared to younger women are almost closer to having completed fertility. Besides, the fact that these women are older and in stable relationships suggest that older women have the necessary experience that equip them to care for additional children. Besides, early sexual activity and childbirth before the age of 18 are associated with higher fertility rates in Zambia, aligning with findings from other SSA [45,46]. This correlation may be attributed to limited access to family planning (FP) information and services among young women, influenced by factors like lack of SRH education [47,48]. Moreover, social and cultural barriers in developing countries can hinder young women's ability to obtain contraception and negotiate safe sex practices with older partners, as condom use maybe perceived negatively [49,50]. These factors can lead to unintended pregnancies, early marriage, and multiple births, ultimately contributing to higher fertility rates and limiting women's educational and economic prospects. This cycle of socio-economic disadvantage can perpetuate poverty and inequality.

Our study revealed that women who preferred more children or were undecided about stopping childbearing had a higher number of children ever born compared to those who wanted no more children or preferred a smaller family size. This finding contrasts with previous research suggesting a lower likelihood of additional births among women who desire family completion no more children have lower likelihood of having more CEB [26,32]. Firstly, women's preferences for family size can evolve post-childbirth, potentially leading to additional pregnancies despite initial intentions [51,52]. Secondly, Zambia's high rate of early marriage (nearly half before age 18) exposes women to extended reproductive periods, contributing to a higher number of CEB [24]. Thirdly, women desiring smaller families may be more likely to use contraception. Limited access to family planning resources could hinder their ability to achieve their preferred or ideal family size [53,54]. Finally, cultural norms in Zambia, which often associate large families with prosperity, social status, and old-age security, may influence women's preferences for larger families [55,56].

The study revealed a correlation between a preference for a specific child's sex and a lower number of children ever born, echoing previous findings [57]. This suggests that social and cultural expectations may influence women to cease childbearing after achieving their desired sex, particularly in cultures valuing sons for lineage or inheritance [58].

Interestingly, the study's finding that married Catholic women in Zambia are less likely to have fewer children than those from other faiths contradicts previous research [59,60]. However, factors like natural family planning, education, healthcare access, economic opportunities, and the emphasis on child well-being may play a more significant role than religion in influencing fertility decisions, even among Catholic women [61]. Additionally, urbanization and associated costs of living may contribute to smaller family sizes, particularly among urban Catholics. Ultimately, a complex interplay of social, cultural, economic, and religious factors shapes women's reproductive choices, with limited education and economic opportunities often hindering access to family planning and reproductive health services.

Socio-economic status also plays a key role in determining the number of CEB among married women in Zambia. Educated women in both developed and less developed countries tend to have fewer children [62,63]. Factors contributing to this include delayed marriage, challenging traditional norms, and career aspirations [64]. Education also empowers women with knowledge about SRH and FP methods, enabling them to make informed choices about family planning [65]. Similarly, our study revealed that women from disadvantaged socio-economic backgrounds and rural areas exhibited higher desired family sizes, a finding corroborated by previous studies [24,66]. While poverty can increase the perceived value of children as labour and social security, it can also limit access to family planning services and resources due to financial constraints and competing priorities [67,68]. Moreover, the burden of child-rearing, particularly in poverty, can influence women to opt for smaller families, however how this is possible in a setting like ours requires further investigation [69]. Therefore, the above demonstrates that the reasons behind demand for more children and poverty are multifaceted, and vary depending on cultural and social contexts.

Studies conducted in SSA indicate that there is a relationship between number of children ever born and woman's decision-making autonomy [29,70]. These findings are similar to findings of this study, revealing that women with greater decision-making control or who believed that wife beating is unjustified have a lower number of CEB. These findings suggest a link between women's empowerment and their sexual and reproductive health choices [26,71] When women have autonomy over their bodies, sexuality, and reproduction, they often exercise greater control over their lives, including family size. Furthermore, empowered women are more likely to effectively use contraception, make informed decisions about childbearing, and reduce unintended pregnancies, and unsafe abortions [71]. Therefore, enhancing women's autonomy is crucial for improved SRH decision-making, empowering women, and promoting their overall health and well-being, including their right to bodily autonomy.

## Study strengths and limitations

This study offers several notable strengths. The representative sampling methodology ensures that the findings can be generalized to married women in Zambia. The analysis of three survey years reveals trends and patterns in fertility behavior. Moreover, this study is a pioneering endeavour in examining the relationship between intermediate variables and fertility within the Zambian context, while simultaneously accounting for a range of bio-demographic and socio-economic factors. This study has two major limitations: the design itself may not account for potential confounders since we used secondary cross-sectional data which limits causation. The fact that we are relying on self-reported data, there are chances that the respondents may have introduced some bias which we cannot control for.

## Conclusion

This study suggests that the two intermediate factors of age at first marriage and abortion are the primary determinants of the number of CEB among married Zambian. Besides, the intermediate factors this study has established that bio-demographic factors such as married women's age, early childbearing, were associated with higher CEB. Surprisingly, women undecided or opposed to more children had higher fertility, indicating a need for further research on childbearing decisions. Furthermore, socio-economic factors like household wealth and residing in rural areas were linked to higher CEB, thereby underlining the complex interplay between economic conditions, cultural expectations, and fertility. Therefore, the findings underscore the need for policies, programs, and interventions addressing early marriage, enhancing sexual and reproductive health service access and utilization, empowering women to make informed fertility decisions while respecting cultural sensitivities to achieve desired fertility goals, and potentially exploring safe abortion services within the legal framework to manage unintended pregnancies. Moreover, further analysis and research should be undertaken to determine whether poverty can lead to reduced fertility in less developed countries of SSA.

## Acknowledgments

We wish to extend our thanks to the Demographic and Health Survey Program for allowing us to use the three datasets for us to undertake this study.

## Author contributions

**Conceptualization:** Bwalya Bupe Bwalya.

**Formal analysis:** Bwalya Bupe Bwalya.

**Methodology:** Bwalya Bupe Bwalya.

**Writing – original draft:** Bwalya Bupe Bwalya, Clifford Odimegwu.

**Writing – review & editing:** Bwalya Bupe Bwalya, Clifford Odimegwu.

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
