## [Decision Letter · Decision Letter 0]

31 Oct 2024

PGPH-D-24-02017

Decoding fertility behaviour of married women in Zambia: A multifaceted analysis of bio-demographic, socio-economic and intermediate factors

Dear Dr. Bwalya,

Thank you for submitting your manuscript to PLOS Global Public Health. After careful consideration, we feel that it has merit but does not fully meet PLOS Global Public Health’s publication criteria as it currently stands. Therefore, we invite you to submit a revised version of the manuscript that addresses the points raised during the review process.

The manuscript has been assessed by two reviewers and their comments are available below and in the attached documents. Please review this comments and make the appropriate revisions to your manuscript. 

We look forward to receiving your revised manuscript.

Kind regards,

Emma Campbell, Ph.D

Staff Editor

Journal Requirements:

Reviewers' comments:

Reviewer's Responses to Questions

**Comments to the Author**

1. Does this manuscript meet PLOS Global Public Health’s publication criteria ? Is the manuscript technically sound, and do the data support the conclusions? The manuscript must describe methodologically and ethically rigorous research with conclusions that are appropriately drawn based on the data presented.

Reviewer #1: Yes

Reviewer #2: Yes

2. Has the statistical analysis been performed appropriately and rigorously?

Reviewer #1: Yes

Reviewer #2: Yes

3. Have the authors made all data underlying the findings in their manuscript fully available (please refer to the Data Availability Statement at the start of the manuscript PDF file)?

Reviewer #1: Yes

Reviewer #2: Yes

4. Is the manuscript presented in an intelligible fashion and written in standard English?

Reviewer #1: Yes

Reviewer #2: Yes

5. Review Comments to the Author

Reviewer #1: The article effectively addresses the subject. However, the following suggestions may help enhance it:

- Consider providing more explanation about intermediate variables in the introduction section.

- The discussion section is too lengthy, with frequent repetition of findings. It could be more concise.

Reviewer #2: 1. The number of children ever born (CEB), as a longitudinal measure, reflects the lifetime fertility of women, unlike the total fertility rate (TFR), which is a cross-sectional measure. Consequently, it is expected that women who married at a younger age (before 18 in this study) or had a history of abortion, compared to those who married after 18 and did not experience an abortion, will exhibit higher fertility. This finding remains significant even after controlling for other explanatory variables.

2. Given the methodological issues associated with Davis and Blake's intermediate fertility variables and the introduction of the proximate determinants (PD) of fertility by John Bongaarts (1978) and Bongaarts and Potter (1983), it is recommended that the authors also refer briefly to the Bongaarts model. Moreover, it is necessary to explain that, considering the nature of the dependent variable (CEB) and its distinction from the total fertility rate (TFR), Davis and Blake's model has been employed as the theoretical framework for this research.

3. In the classification of control variables, I observe some interference in how the variables are grouped. For instance, sexual preference, religious affiliation, and the decision-making autonomy of women should be classified as cultural variables. Additionally, place of residence is a characteristic related to population distribution, and I propose that it be included among the demographic variables.

4. The variable of religious affiliation is divided into two categories: Catholic and others. It is necessary to specify which religious groups are included in the 'others' category.

5. This article explains fertility in Zambia, eliminating the need to repeat "Zambia" in some titles, such as the following:

•Bio-demographic, socio-economic, and intermediate correlates of children ever born in Zambia.

•Factors associated with children ever born in Zambia

6. Typing and writing errors are evident throughout the article, some highlighted in the PDF file. It is essential to edit the article carefully once more.

7. The analysis on page18 lacks theoretical support unless specific reasons are provided:

“On the contrary, women being poor can also make it difficult to afford raising children, with limited access to proper nutrition, healthcare, and education [68]. This can lead some women to choose to have fewer children. Therefore, the above demonstrates that the reasons behind demand for more children and poverty are multifaceted, and vary depending on cultural and social contexts.”

Typically, families living in poverty have a higher desire for children and an ideal number of children than those in better living conditions, as wealth flows from children to parents in poorer households. Further analysis is required.

6. PLOS authors have the option to publish the peer review history of their article (what does this mean? ). If published, this will include your full peer review and any attached files.

**Do you want your identity to be public for this peer review?** For information about this choice, including consent withdrawal, please see our Privacy Policy .

Reviewer #1: No

Reviewer #2: **Yes: ** Hatam Hosseini

---

## [Decision Letter · Decision Letter 1]

2 Dec 2024

Decoding fertility behaviour of married women in Zambia: A multipronged analysis of bio-demographic, socio-economic and intermediate factors

PGPH-D-24-02017R1

Dear Mr. Bwalya,

We are pleased to inform you that your manuscript 'Decoding fertility behaviour of married women in Zambia: A multipronged analysis of bio-demographic, socio-economic and intermediate factors' has been provisionally accepted for publication in PLOS Global Public Health.

Best regards,

Julia Robinson

Executive Editor

Reviewer Comments (if any, and for reference):

Reviewer's Responses to Questions

**Comments to the Author**

1. If the authors have adequately addressed your comments raised in a previous round of review and you feel that this manuscript is now acceptable for publication, you may indicate that here to bypass the “Comments to the Author” section, enter your conflict of interest statement in the “Confidential to Editor” section, and submit your "Accept" recommendation.

Reviewer #1: (No Response)

Reviewer #2: All comments have been addressed

2. Does this manuscript meet PLOS Global Public Health’s publication criteria ? Is the manuscript technically sound, and do the data support the conclusions? The manuscript must describe methodologically and ethically rigorous research with conclusions that are appropriately drawn based on the data presented.

Reviewer #1: (No Response)

Reviewer #2: Yes

3. Has the statistical analysis been performed appropriately and rigorously?

Reviewer #1: (No Response)

Reviewer #2: Yes

4. Have the authors made all data underlying the findings in their manuscript fully available (please refer to the Data Availability Statement at the start of the manuscript PDF file)?

Reviewer #1: (No Response)

Reviewer #2: (No Response)

5. Is the manuscript presented in an intelligible fashion and written in standard English?

Reviewer #1: (No Response)

Reviewer #2: Yes

6. Review Comments to the Author

Reviewer #1: no comments

Reviewer #2: (No Response)

7. PLOS authors have the option to publish the peer review history of their article (what does this mean? ). If published, this will include your full peer review and any attached files.

**Do you want your identity to be public for this peer review?** For information about this choice, including consent withdrawal, please see our Privacy Policy .

Reviewer #1: No

Reviewer #2: No
